# EPI Distortion Correction without Opposite Phase Encodings with Unsupervised INR-Based Deformable Registration

**Tyler Spears**[1] (iD)                                           TAS6HH@VIRGINIA.EDU
[1] *Dept. of Electrical & Computer Engineering, University of Virginia, Charlottesville, VA, USA*

**Myla Goldman**[2] (iD)                                          MYLA.GOLDMAN@VCUHEALTH.ORG
[2] *Dept. of Neurology, Virginia Commonwealth University School of Medicine, Richmond, VA, USA*

**P. Thomas Fletcher**[1] (iD)                                    PTF8V@VIRGINIA.EDU

**Editors:** Accepted for publication at MIDL 2026

## Abstract

Diffusion MRIs (dMRIs) provide a detailed look at the structure of the brain, but the acquired images come with many distortions. Echo planar imaging (EPI) sequences, nearly universal for dMRIs, are highly sensitive to inhomogeneities of the magnetic field in the scanner. This results in severe geometric distortion (up to tens of millimeters) in the phase encoding direction, particularly in areas with strong changes in tissue density such as the brainstem, temporal, and frontal regions. A common method for correcting EPI distortion is to collect an image with the opposite phase encoding (PE) direction and reconstruct the magnetic susceptibility field. However, many dMRI protocols, some still in use today, do not include this auxiliary acquisition. Other methods have attempted to register the distorted EPI to an anatomical reference, with less accurate results. In this work, we propose EPINR, an unsupervised implicit neural representation (INR) based registration model that builds on these previous works. EPINR learns the susceptibility field by warping a single b0 image to a T1w reference, without opposite PE acquisitions. EPINR also leverages its smooth and continuous representation to apply higher-order regularizations calculated analytically. We evaluate EPINR against several comparison methods, both traditional and learning-based, over two dMRI datasets. We perform further ablation analyses on the effect of different components in EPINR. Finally, we discuss the reasons for EPINR's high performance, and how it can bring structural precision to previously compromised diffusion images.

**Keywords:** Intermodal Registration, INR, Susceptibility Distortion Correction

## 1. Introduction

Diffusion MRI (dMRI) is the most detailed structural description of the brain we can acquire *in vivo*. Unfortunately, that detail is compromised by severe geometric distortions, on the order of tens of millimeters, caused by magnetic field inhomogeneities in the scanner (Farzaneh et al., 1990; Haskell et al., 2023). These distortions affect nearly all dMRI scan sequences, and field inhomogeneities are produced whenever any tissue or material is in the scanner. So, nearly all dMRIs need this distortion, often called susceptibility distortion or echo planar image (EPI) distortion, to be corrected.

Several susceptibility distortion correction (SDC) methods have been previously proposed, but the most effective methods require auxiliary acquisitions explicitly for SDC. For example, the popular FSL `topup` (Andersson et al., 2003) tool performs SDC with impressive accuracy (Gu and Eklund, 2019; Graham et al., 2017; Wang et al., 2017), and is a staple in dMRI processing pipelines (Glasser et al., 2013; Cieslak et al., 2021). The `topup` tool requires only a single b0 collected with the opposite phase encoding (PE) direction as the primary diffusion weighted images (DWIs). However, that single image is not collected in many dMRI datasets, especially in places that rely on older sequences. Examples include: the Open Access Series of Imaging Studies 3 (OASIS3) dataset (LaMontagne et al., 2019), the Center on Reproducible Research (CoRR) dataset (Zuo et al., 2014), the MASiVar multisite and multiscanner dataset (Cai et al., 2021), the TractoInferno dataset (Poulin et al., 2021), and many other public (and private) dMRI datasets.

Alternative SDC methods have been proposed that only require an undistorted anatomical reference image, such as a T1-weighted (T1w) or T2-weighted (T2w) image, which are almost always collected in dMRI protocols (Kybic et al., 2000). However, these deformable registration-based methods are significantly less accurate than methods, such as `topup`, that require auxiliary images (Gu and Eklund, 2019; Graham et al., 2017; Wang et al., 2017). New unsupervised implicit neural representation (INR) (Sitzmann et al., 2020) registration methods (Wolterink et al., 2022; Byra et al., 2023) could bring new life into this decades-old problem of SDC without extra acquisitions.

In this work, we propose EPINR, an unsupervised INR-based deformable registration model for EPI distortion correction without extra acquisitions. EPINR learns a smooth displacement field, constrained to the PE direction, that aligns a b0 and T1w image pair for a particular subject. EPINR also leverages the spatially-continuous and smooth nature of INRs to *analytically* compute Jacobians and higher-order derivatives for regularization (Rueckert et al., 1999; Wolterink et al., 2022). Our contributions in this work are as follows:

- EPINR, an unsupervised INR-based registration model that is the first of its kind to be applied to SDC without auxiliary acquisitions.

- A comparison of previous methods, traditional and deep learning-based, on two DWI datasets with vastly different image characteristics, including an in-house dataset not used in any previous work on SDC validation.

- Validation that EPINR outperforms previous methods, and an analysis of where a previous state of the art method struggles.

- An ablation analysis on the effects of regularization schemes and domain-specific enhancements of EPINR.

- An open source implementation of EPINR at https://github.com/TylerSpears/epinr.

## 2. Background

EPI distortions cause a strong geometric warping during the data readout step of an EPI sequence, which effects both dMRIs and functional MRIs (fMRIs) (Studholme et al., 2000;

Gholipour et al., 2006). This is caused by inhomogeneities in the magnetic susceptibility field in the scanner, causing a non-linear shift in the scanner's gradient field at tissue boundaries (Johansen-Berg and Behrens, 2014, Chapter 4). This distortion is almost entirely constrained to the phase encoding (PE) direction, and it is most severe in the sinuses and the brainstem, while also affecting temporal and frontal regions (Treiber et al., 2016). We also note that for dMRIs, SDC is usually performed only on images with no diffusion weighting (b0s), as they do not contain eddy current-induced distortions (Andersson et al., 2017; Rohde et al., 2004).

## 2.1. SDC with Extra Acquisitions

As mentioned, SDC methods that leverage auxiliary acquisitions often reconstruct the most accurate susceptibility fields (Gu and Eklund, 2019; Graham et al., 2017; Wang et al., 2017). This includes fieldmap-based SDC and reverse PE SDC. Some methods suggested collecting fieldmaps that measure the magnetic inhomogeneity of the subject (Jezzard and Balaban, 1995; Wan et al., 1997; Reber et al., 1998), similar to those in fMRI, but comparisons (Tao et al., 2009; Fritz et al., 2014; Wang et al., 2017) have found that these approaches are sensitive to acquisition parameters and partial volume effects. Reverse PE SDC is more common, where a b0 image with an opposite PE direction is collected. This only adds a few seconds to the scan time, but is still absent in many datasets and sequences. That said, reverse PE images are sufficiently common, and their performance is very high, such that they are raised to a "silver standard" in SDC (Gu and Eklund, 2019). These methods assume that images with opposite PEs will have an equal and opposite displacement field, and a parameterized field is optimized with these constraints.

Reverse PE SDC was first proposed by Chang and Fitzpatrick (1992) and popularized with the FSL `topup` tool (Andersson et al., 2003). Other methods of this class continue to be developed, such as those in Holland et al. (2010), Irfanoglu et al. (2015) (DR-BUDDI), Hédouin et al. (2017), and Liu et al. (2021). Deep learning has also been applied, with EPIs of both PE directions given to a convolutional neural network (CNN) for predicting the susceptibility field (Hu et al., 2020; Qiu et al., 2026; Zahneisen et al., 2020; Legouhy et al., 2022; Qiao and Shi, 2022; Zaid Alkilani et al., 2024). We also mention that pulse sequence design for reducing EPI distortion is an active area of research (Haskell et al., 2023), but these sequences are still experimental and cannot help with post hoc analyses.

## 2.2. Deep Learning Deformable Registration

**VoxelMorph & CNN Registration.** Deep learning-based registration methods have exploded in popularity after the popular VoxelMorph framework (Balakrishnan et al., 2019) was proposed. VoxelMorph uses a CNN that takes the fixed and moving images as inputs and predicts a deformation field that warps the moving image to the fixed. The primary objective function is an image similarity measure, but also includes regularization terms that penalize the squared norm of the displacement spatial gradient. Note that this gradient is approximated with a finite difference method.

**INR-Based Registration.** INRs, first proposed as sinusoidal representation networks (SIRENs) in Sitzmann et al. (2020), are multilayer perceptrons (MLPs) that learn a continuous signal representation by mapping coordinates to intensity values. INR registration

represents a warp field by using an image similarity objective function, rather than image reconstruction (Wolterink et al., 2022). If properly constructed, this representation is $C^\infty$ smooth, and may analytically calculate higher order derivatives and use advanced regularization techniques (Rueckert et al., 1999). Wolterink et al. (2022) was one of the first of these methods with implicit deformable image registration (IDIR). Sun et al. (2024) proposed the Neural Image Registration (NIR) model, which extended IDIR with a hybrid coordinate sampling scheme, and van Harten et al. (2024) proposed a cycle-consistent deformation with dual INRs. Byra et al. (2023) analyzed the impact of different INRs on deformable brain image registration.

### 2.3. Previous Works in SDC Without Auxiliary Acquisitions

**Traditional Registration.** A third class of SDC method is to register the distorted b0 to an undistorted anatomical reference, constrained to the PE direction. While this problem may seem straightforward, the intermodal differences between a b0 (T2-weighted) and a T1w, along with the difference in noise levels and resolution, prove challenging. Kybic et al. (2000) was the first to propose a deformable registration SDC method. Later, Gholipour et al. (2006) proposed a Free-Form Deformation method for fMRIs, and Tao et al. (2009) used a variational approach to register the b0 with a *T2w* reference. However, one of the most common models for this approach to apply a general image registration framework, such as ANTs symmetric normalization (SyN) (Avants et al., 2008) to the task, as shown in Wang et al. (2017); Gu and Eklund (2019). Unfortunately, registration-based correction, while requiring no extra acquisitions, also gives the lowest performance of the three SDC methods (Gu and Eklund, 2019; Graham et al., 2017; Wang et al., 2017).

**Learning-Based Registration** The availability of large DWI repositories has inspired applications of neural networks to EPI distortion correction. As mentioned previously, many works have used the opposite PE approach to predict the susceptibility fields, but few have attempted SDC without auxiliary images. Ye et al. (2021) proposed a model to perform simultaneous super-resolution and distortion correction with only one PE direction. Jimeno et al. (2024) proposed GDCNet, which applies VoxelMorph to EPI distortion in fMRIs, but has not yet been peer-reviewed at the time of this writing. Possibly the most popular deep network SDC method is Synb0 (Schilling et al., 2019), which is a conditional generative adversarial network (GAN) that synthesizes an undistorted b0 when given a distorted b0 and a T1. This method was trained in 2.5D (2D slices in all three axes) on a dataset of over 500 subjects, with output images at 2.5mm isotropic. The real and generated b0 images are then given to `topup` for SDC. This model was used for SDC for datasets in Begnoche et al. (2022), Poulin et al. (2021), Cai et al. (2021), and more.

## 3. EPINR

Our goal with EPINR is to correct for EPI distortion in a diffusion image protocol using only a single b0 (effected by EPI distortion) and a T1-weighted anatomical reference (not effected by this distortion), and our model is illustrated in Figure 1. The underlying susceptibility field can be calculated simply by scaling a displacement field $u(x, y, z)$, so registration and field reconstruction are the same problem here. Like classical registration models, we train EPINR in an unsupervised manner with a newly initialized model for every subject.

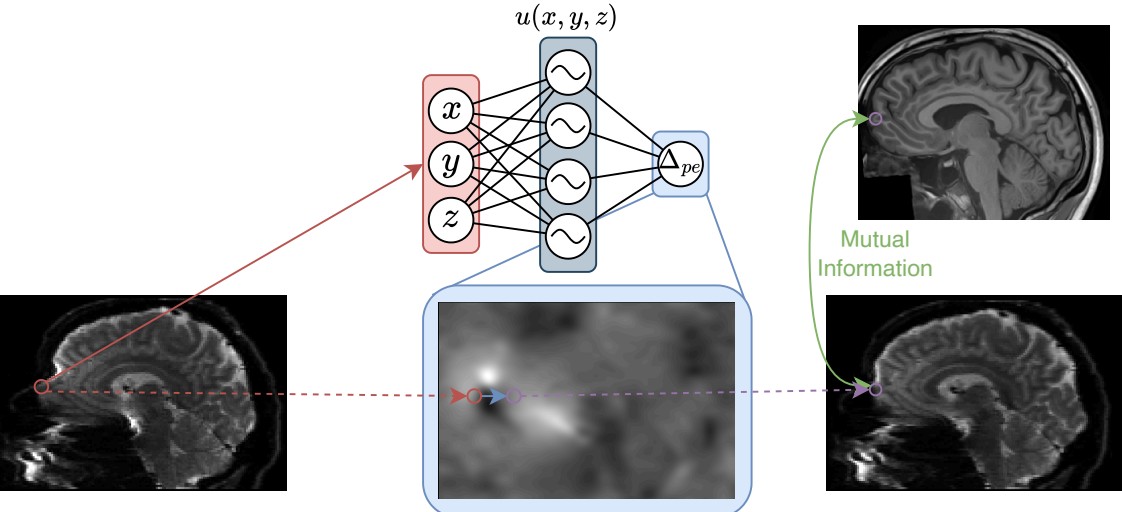

Figure 1: Illustration of our proposed EPINR registration model. EPINR learns to warp a distorted b0 to a non-distorted T1w reference in an unsupervised manner with a displacement field (blue) constrained in the phase encoding direction. The $C^\infty$ smoothness of EPINR also allows for *analytically* calculating higher-order regularization methods and Jacobian modulation (Jezzard and Balaban, 1995).

Suppose we have a spin echo pulse sequence (Johansen-Berg and Behrens, 2014, Chapter 2) that produces a b0 image $\mathbf{M}$ distorted only in the PE direction, and a non-distorted T1w image $\mathbf{F}$. We assume, without loss of generality, that the PE direction is $y$. Additionally, suppose that as a preprocessing step, $\mathbf{F}$ has been rigidly aligned to match the orientation and translation of $\mathbf{M}$. Note that we keep $\mathbf{F}$ and $\mathbf{M}$ in their native resolutions as we will just sample locations in $\mathbf{F}$ and $\mathbf{M}$ during training. We denote this sampling operation as $\mathbf{A}[\mathbf{c}]$, which is a trilinear interpolation of volume $\mathbf{A}$ at the 3D physical coordinate $\mathbf{c}$. So, our network, a SIREN (Sitzmann et al., 2020) in our implementation, $u_\theta : \mathbb{R}^3 \to \mathbb{R}$ with parameters $\theta$, maps $u_\theta(x_i, y_i, z_i) = (0, \Delta y_i, 0)$, with zeros in the non-PE directions.

It is known that EPI distortion in spin echo sequences is mass preserving (Chang and Fitzpatrick, 1992; Jezzard and Balaban, 1995) in that compressed signal is "piled-up" to a higher intensity, and vice-versa for "stretched" areas. This is expressed as a division of the image intensity by the determinant of the Jacobian of the deformation field, and we need to compensate for that when applying our predicted displacement field. This is also known as Jacobian modulation (Andersson et al., 2003). So, our similarity loss is defined as:

$$\mathcal{L}_{\text{sim}} = \frac{1}{N} \sum_{i=1}^{N} -\text{MI} \left( \det J_u \left( \mathbf{c}_i + u_\theta(\mathbf{c}_i) \right) \mathbf{M} \left[ \mathbf{c}_i + u_\theta(\mathbf{c}_i) \right], \mathbf{F} \left[ \mathbf{c}_i \right] \right), \tag{1}$$

where $\mathbf{c}_i$ is a patch of contiguous coordinates in the $i$'th batch, $J_u$ is the Jacobian of the deformation field produced by the EPINR network $u$, and MI is the mutual information

similarity (Viola and Wells III, 1997). The det $J_u$ term applies Jacobian modulation, but this modulation is changing along with the network weights during training. This complicates the optimization process (as mentioned in Andersson et al. (2001); Tao et al. (2009)), but potentially increases final registration accuracy as modulation will always be applied to the final undistorted b0. We also reiterate that $J_u$ is analytically calculated on the network parameters $\theta$, not with a finite difference approximation (Wolterink et al., 2022).

Taking inspiration from the SDC method in fmriprep (Esteban et al., 2019), we also added a similarity term for the Laplacians of the $\mathbf{F}$ and the undistorted $\mathbf{M}$:

$$\mathcal{L}_{\text{lap}} = \frac{1}{N} \sum_{i=1}^{N} -\text{NCC}\left(\nabla^2\left(\det J_u\left(\mathbf{c}_i + u_\theta(\mathbf{c}_i)\right)\mathbf{M}\left[\mathbf{c}_i + u_\theta(\mathbf{c}_i)\right]\right), \nabla^2(1 - \mathbf{F}\left[\mathbf{c}_i\right])\right), \quad (2)$$

where NCC is normalized cross-correlation, $\nabla^2$ is the image Laplacian operator, and $1 - \mathbf{F}$ flips the T1w intensity distribution to more closely match the b0.

Finally, we used two regularization functions: a smoothing loss as the Frobenius norm of the Jacobian, and a bending energy loss that requires the Hessian of the deformation (Rueckert et al., 1999). These are given as:

$$\mathcal{L}_{\text{smooth}} = \|J_u(\mathbf{c}_i + u_\theta(\mathbf{c}_i))\|_F = \left|1 + \frac{\partial u_\theta(\mathbf{c}_i)}{\partial y}\right| \quad (3)$$

$$\mathcal{L}_{\text{bend}} = \sum_{p=1}^{3}\sum_{q=1}^{3}\sum_{r=1}^{3}\left(\mathbf{H}_p^{(u)}(\mathbf{c}_i + u_\theta(\mathbf{c}_i))\right)_{q,r}^2 = \sum_{q=1}^{3}\sum_{r=1}^{3}\left(\mathbf{H}_{p=2}^{(u)}(\mathbf{c}_i + u_\theta(\mathbf{c}_i))\right)_{q,r}^2, \quad (4)$$

with simplified forms on the right side enabled by the constraint to the PE direction. Here, $(\mathbf{H}_p^{(u)}(\mathbf{c}_i + u_\theta(\mathbf{c}_i))_{q,r}$ is the analytically calculated Hessian matrix of the deformation field produced by the network $u_\theta$ at the $p$'th output dimension with relation to input dimensions $q$ and $r$ (where $x = 1$, $y = 2$, and $z = 3$). Together with their corresponding weight terms, these four loss terms form our full objective function.

## 4. Experiments & Results

### 4.1. Data

We evaluated EPINR and the comparison methods on two different datasets. All methods were either unsupervised or trained on other datasets, so there were no train/test splits. All evaluated b0s were acquired with an anterior to posterior (AP) PE direction, and only one b0 was used. All DWIs were denoised with Marchenko-Pastur principal component analysis (MP-PCA) (Veraart et al., 2016) and had $B_1$ bias field correction applied with N4 (Tustison et al., 2010). For topup, multiple (two for MICA-MICS, three for UVA MS) AP b0s and PA b0s were used for SDC. T1w images were bias corrected with N4 and denoised with a non-local means filter (Manjón et al., 2010). Then, T1w images were rigidly registered to the *distorted* b0s with ANTs neighborhood normalized CC (Avants et al., 2008). Tissue masks were estimated with the FSL brain extraction tool (BET) (b0) (Smith, 2002) and SynthStrip (T1) (Hoopes et al., 2022). These masks weighed voxel contributions in training.

The first dataset was the publicly available Microstructure-Informed Connectomics (MICA-MICS) dataset (Royer et al., 2021) with $N = 49$ subjects. DWIs were collected with a spin-echo sequence at 1.6mm isotropic resolution, TR=3500ms, TE=64.40ms, and an FSL-style

total readout time of 0.05282 seconds. T1w images were acquired with an MP-RAGE sequence, 0.8mm isotropic resolution, TR=2300ms, and TE=3.14ms. The second dataset was an in-house DWI dataset collected at Virginia Commonwealth University (VCU) as part of a study involving multiple sclerosis (MS) (Pearsall et al., 2024; Goldman et al., 2023), which we name "VCU MS." This study was approved by the VCU Institutional Review Board. We evaluated methods on $N = 48$ patients from VCU MS, including 10 healthy controls and 38 patients diagnosed with MS. DWIs were acquired at 1.875mm in-plane and 2mm slice thickness, TR=4900ms, TE=118ms, and a total readout time of 0.063054 seconds. The T1w images were acquired at 1mm isotropic resolution with TE=2.99ms.

### 4.2. Implementation Details

We implemented EPINR as a SIREN (Sitzmann et al., 2020) MLP with 256 hidden features and 5 layers. Training was performed on randomly sampled patches as a full b0 could not fit into the graphical processing unit (GPU) memory, effective batch size of 6, 50 batches per epoch, over 60 epochs. Adam with weight decay (Loshchilov and Hutter, 2018) was used for optimization, with an initial learning rate of $5 \times 10^{-4}$ that linearly decreased to $5 \times 10^{-6}$ with a small number of warmup and cooldown epochs. Both the b0 and T1w volumes were normalized to $[0, 1]$. The input coordinates $\mathbf{c}_i$ were normalized to $[-1, 1]$ based on the b0's field of view, and MLP outputs were re-scaled to physical units. The image Laplacian $\nabla^2$ in Equation 2 was implemented as a difference of Gaussian filters (Marr and Hildreth, 1980).

### 4.3. Comparison Methods

We compare EPINR to several SDC methods on both datasets:

1. *EPINR* - Our proposed model that uses an INR to learn the displacement field between a b0 and a T1w image, without resizing either out of their native resolutions.

2. *Uncorrected* - The acquired b0 with preprocessing and no SDC applied.

3. *ANTs-SyN* - ANTs Syn deformable registration (Avants et al., 2008) with parameters from the `antsIntermodalityIntrasubject.sh` script with PE direction constraints.

4. *QSIPrep* - Fieldmapless SDC built into the QSIPrep dMRI preprocessing pipeline (Cieslak et al., 2021), version 1.0.2. This is tweaked variant of ANTs-SyN (Avants et al., 2008) with custom preprocessing; this is given as "experimental" within QSIPrep.

5. *Synb0+Topup* - Synb0 synthesized undistorted b0 volume with follow-up SDC with topup (Schilling et al., 2019). Synb0 uses a trained GAN to synthesize an undistorted b0 at 2.5mm isotropic resolution. The preprocessed b0s and T1s were masked for input, using the scilpy (Boré et al., 2025) wrapper Synb0. The synthetic b0 is upsampled to the b0's native resolution, and topup performs SDC.

6. *Topup GT* - FSL topup (Andersson et al., 2003) run with b0s that have AP and PA PE directions, considered as our ground truth.

For our ablation analysis, we compared the full EPINR model with several variants: *No Regular.*, which removed both regularization terms $\mathcal{L}_{\text{smooth}}$ and $\mathcal{L}_{\text{bend}}$; *No Bend. Regular.*, which removed bending energy regularization term $\mathcal{L}_{\text{bend}}$; *No Lapl. Sim.*, where Laplacian similarity $\mathcal{L}_{\text{lap}}$ was removed; *No Jac. Mod.*, which disabled Jacobian modulation during training, but is still applied after training; and *EPINR-128x3*, a reduced EPINR model with 3 hidden layers with 128 features.

## 4.4. Evaluation Metrics

We transform each model's predicted displacement field (in mm) to a susceptibility field (Hz) with scaling by the PE direction resolution and total readout time. Then, we use the `applytopup` FSL tool to apply the susceptibility field to the distorted b0 with Jacobian modulation. We then measure each model's performance over five metrics: 1) a mean-squared error (MSE) of the model's undistorted b0 vs. topup's undistorted b0, 2) MSE of the model's predicted susceptibility field vs. topup's reconstructed field, 3) MI similarity between the model's corrected b0 and the T1w reference, with both normalized to $[0, 1]$, 4) local normalized CC (LNCC) of the corrected b0 and the T1w reference image, and 5) percentage of negative voxels in the determinant of the Jacobian of the model's deformation field, calculated with a central difference method to keep evaluation consistent.

## 4.5. Results

| Dataset | Model | b0-topup MSE $\times 10^{-6}$ ↓ | SF-topup MSE $\times 10^{-3}$ ↓ | MI b0-Anat. ↑ | LNCC b0-Anat. ↑ |
|---|---|---|---|---|---|
| MICA-MICS N=49 | Uncorrected | 1.3021 (0.300) | 0.5397 (0.125) | 0.3675 (0.034) | 0.2271 (0.014) |
| | ANTs-SyN | 1.6665 (0.354) | 0.5229 (0.092) | 0.3342 (0.054) | 0.2395 (0.017) |
| | QSIPrep | 1.4983 (0.361) | 0.5710 (0.128) | 0.3677 (0.033) | 0.2252 (0.012) |
| | Synb0+Topup | 1.3741 (0.338) | 1.4491 (0.267) | 0.4285 (0.034) | 0.2138 (0.013) |
| | EPINR | **0.9878 (0.237)** | **0.4221 (0.087)** | **0.4299 (0.051)** | **0.2501 (0.016)** |
| | Topup GT | - | - | 0.4629 (0.039) | 0.2639 (0.019) |
| VCU MS N=48 | Uncorrected | 4.1649 (1.375) | 0.8881 (0.248) | 0.2750 (0.040) | 0.1621 (0.016) |
| | ANTs-SyN | 3.5424 (1.448) | 0.7522 (0.189) | 0.2167 (0.039) | 0.1840 (0.016) |
| | QSIPrep | 3.2523 (1.066) | 0.8075 (0.222) | 0.2667 (0.038) | 0.1720 (0.017) |
| | Synb0+Topup | 3.3660 (1.324) | 2.7373 (0.900) | 0.3276 (0.023) | 0.1540 (0.016) |
| | EPINR | **2.2992 (1.798)** | **0.6569 (0.526)** | **0.3690 (0.031)** | **0.1877 (0.018)** |
| | Topup GT | - | - | 0.3146 (0.027) | 0.1780 (0.020) |

Table 1: Results of different EPI distortion correction methods applied to two DWI datasets. Metric name arrows indicate the direction of better performance, and the best performance between all models is bolded.

Quantitative results for comparing all datasets, models, and metrics (except the Jacobian negative percentage) are shown in Table 1, and examples from both datasets are shown in Figure 2. For the negative Jacobian percentage, the large majority of methods showed 0 negative voxels, except for: Synb0+topup on MICA-MICS (0.0152%), Synb0+topup on VCU MS (0.0017%), and regular topup on VCU MS (0.0006%).

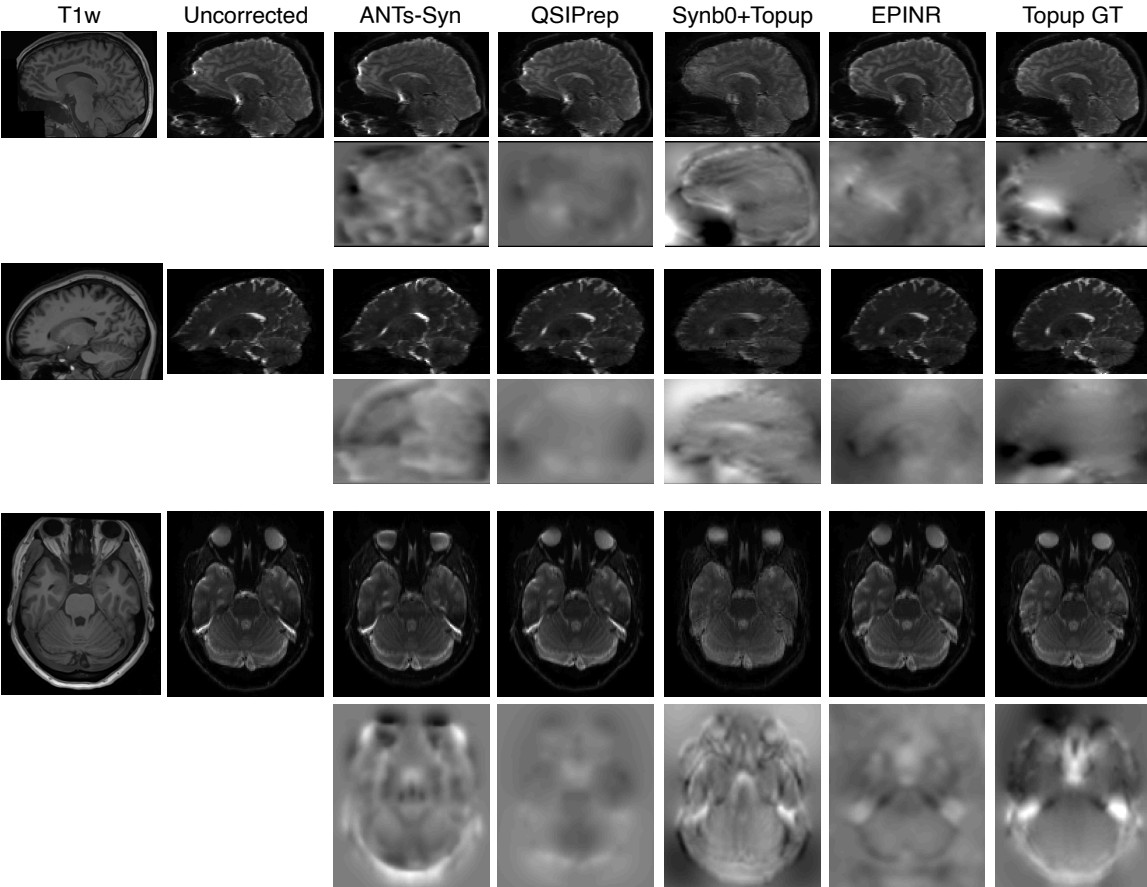

Figure 2: Qualitative results of susceptibility distortion correction over three subjects. Below each undistorted b0, the predicted susceptibility field in Hz is displayed. From top to bottom, subjects are from MICA-MICS, VCU MS, and MICA-MICS.

As shown in Table 1, EPINR outperforms comparison methods on every metric we evaluated over both datasets except for b0 MSE in the MICA-MCIS data, where the ablation model performed the best. We believe that this overall high performance may be due to the high number of parameters that allows EPINR to learn high-frequency warps. According to Figure 2 rows 1 and 3, we also see how including the Jacobian modulation in training helps EPINR "break through" some of the more severe warping in frontal areas, compared to ANTs-SyN and QSIPrep. The 0 negative Jacobian voxels also indicate that the smooth and bending energy regularizers are effective at eliminating "folds" in the resulting deformation fields, in agreement with previous work in Wolterink et al. (2022).

The performance of Synb0 is somewhat surprising. Despite its use in several publicly released datasets, Synb0 performed the worst on some of our metrics. The predicted susceptibility fields in Figure 2 indicate that the topup registration to the synthesized b0 is a poor fit. This may be explained by the fixed spatial resolution of 2.5mm isotropic generated by Synb0, which must be upsampled to 1.6mm for MICA-MICS and 1.875mm for VCU MS and may be too large of a resolution gap for topup to handle. We also found that the generated b0s "filled out" brain tissue into the outer surface of the skull, which changed the basic shape and volume of the b0. This may be caused by an unexpected brain mask shape resulting from our preprocessing, but this would imply that Synb0 is overly sensitive to how masking is performed, which is a difficult task in distorted images.

The results also reveal the limitations of calculating image similarity between the b0 and T1w reference. In the VCU MS data, EPINR "outperforms" topup on image similarity for both MI and LNCC. However, a qualitative comparison of topup and EPINR still confirms that topup produces a better undistorted b0. This discrepency is likely due to 1) EPINR's much higher number of parameters allowing it to somewhat overfit to the anatomical reference, and 2) the simple fact that image similarity between two modalities is not on its own sufficient to measure SDC performance. So, while b0-T1w similarity is often reported in the literature, interpreting results must be done carefully.

## 4.6. Ablation Results

To understand the different components of EPINR, we also performed an ablation analysis with subsampled datasets, where 10 subjects were randomly selected for both MICA-MICS and VCU-MS. The ablated EPINR variants are listed in Section 4.3, and all training and evaluation configurations are the same as Section 4.5.

The quantitative results for our ablation experiment are found in Table 2, and example images are shown in Figure 3. Most models have 0% negative Jacobian determinant voxels, except for the "No Regular." model which had 0.230% and 0.535% negative Jacobian voxels for MICA-MICS and VCU MS, respectively. We find that overall, all included loss terms and the parameters in EPINR all contribute to its performance in different ways. For $\mathcal{L}_{lap}$ under "No Lapl. Sim.," we found that removing this term reduced EPINR's performance slightly in most metrics, although the differences are subtle qualitatively (Figure 3 columns 1 and 4). Specifically, we observe that the Laplacian similarity helps the model from displacing too far out of a T1w tissue boundary. Similarly, we found that the smaller 128x3 MLP ("EPINR-128x3") reduced the model performance in most metrics due to the inability of the smaller network in capturing high frequencies.

| Dataset | Model | b0-topup MSE $\times 10^{-6}$ ↓ | SF-topup MSE $\times 10^{-3}$ ↓ | MI b0-Anat. ↑ | LNCC b0-Anat. ↑ |
|---|---|---|---|---|---|
| MICA-MICS N=10 | Uncorrected | 1.3488 (0.259) | 0.5784 (0.135) | 0.3849 (0.038) | 0.2304 (0.018) |
| | No Regular. | 1.8039 (0.399) | 1.0106 (0.251) | **0.4575 (0.077)** | 0.2539 (0.020) |
| | No Lapl. Sim. | 1.0700 (0.246) | 0.4942 (0.133) | 0.4500 (0.061) | 0.2514 (0.019) |
| | No Bend. Regular. | 1.0214 (0.238) | 0.4778 (0.124) | 0.4491 (0.058) | **0.2547 (0.020)** |
| | No Jac. Mod. | **0.9232 (0.156)** | **0.4315 (0.082)** | 0.4069 (0.040) | 0.2519 (0.021) |
| | EPINR-128x3 | 0.9577 (0.238) | 0.5112 (0.140) | 0.4400 (0.055) | 0.2494 (0.019) |
| | EPINR | 1.0054 (0.220) | 0.4710 (0.124) | 0.4466 (0.063) | 0.2543 (0.020) |
| | Topup GT | - | - | 0.4835 (0.045) | 0.2681 (0.021) |
| VCU MS N=10 | Uncorrected | 4.1100 (1.229) | 0.8621 (0.298) | 0.2680 (0.040) | 0.1627 (0.013) |
| | No Regular. | 2.8701 (1.582) | 1.5217 (0.933) | **0.3886 (0.040)** | 0.1860 (0.021) |
| | No Lapl. Sim. | 2.6543 (1.200) | 0.7439 (0.475) | 0.3699 (0.036) | 0.1853 (0.017) |
| | No Bend. Regular. | **2.6021 (1.238)** | **0.7357 (0.529)** | 0.3713 (0.036) | **0.1882 (0.017)** |
| | No Jac. Mod. | 3.2984 (1.424) | 0.8411 (0.563) | 0.2655 (0.040) | 0.1821 (0.017) |
| | EPINR-128x3 | 2.6682 (1.249) | 0.7803 (0.560) | 0.3546 (0.032) | 0.1847 (0.017) |
| | EPINR | 2.6046 (1.247) | 0.7400 (0.540) | 0.3712 (0.037) | 0.1878 (0.017) |
| | Topup GT | - | - | 0.3057 (0.024) | 0.1760 (0.013) |

Table 2: Ablation results for EPINR models applied to 20 randomly selected subjects (10 from each dataset). Arrows indicate the direction of better performance, the best score is bolded, and the second best is underlined.

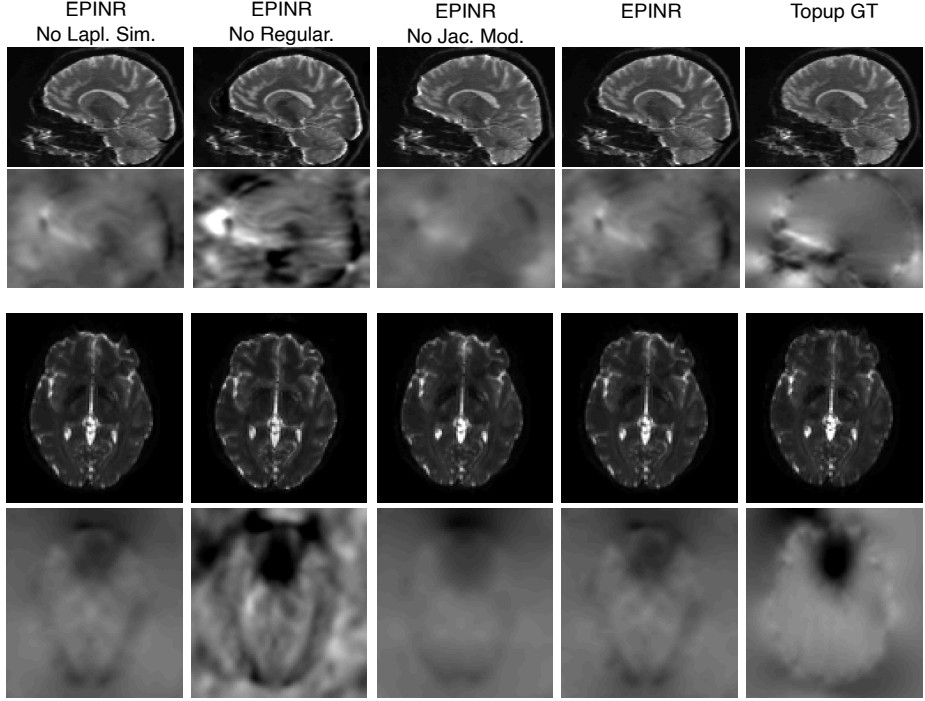

Figure 3: Qualitative results of the ablation experiment. Examples from the MICA-MICS and VCU MS are shown in the top and bottom rows, respectively. Undistorted b0s and are shown above their respective susceptibility field maps.

For our regularization terms $\mathcal{L}_{\text{smooth}}$ and $\mathcal{L}_{\text{bend}}$, we found that some regularization is necessary to prevent invalid deformations. As discussed in Section 4.5, removing regularization causes a high MI similarity, but qualitative inspection in Figure 3 indicates that this is the model overfitting to the T1w. The bending energy term $\mathcal{L}_{\text{bend}}$ specifically was helpful in MICA-MICS, but may have caused a slight performance hit in VCU MS.

The impact of using Jacobian modulation during training is mixed, depending on the dataset. Training with the modulation enabled reduces performance in MICA-MICS, but raises performance in UVA MS. We hypothesize that this depends on the contrast between white and gray matter in the b0 images, where the higher contrast in MICA-MICS leads to the model "overrusing" the Jacobian modulation to match the contrast levels in the T1w image. However, in VCU MS, the contrast between gray and white matter is very low. As shown in Figure 3, columns 3 and 4, we still see that Jacobian modulation allows EPINR to properly decompress severe distortions, but at a higher risk of T1w overfitting.

## 5. Discussion

We have proposed EPINR, a novel INR-based unsupervised registration method for correcting EPI distortions in dMRIs without auxiliary acquisitions. We compared EPINR to several models currently in use, and found that EPINR produced higher quality susceptibility field reconstructions. We validated EPINR's flexibility over two dMRI datasets with different resolutions and image characteristics, and we performed an ablation analysis to better understand the effect of each component in EPINR.

We believe that EPINR's flexibility lends itself to further extensions. One more traditional extension is in utilizing multiple acquisitions or modalities when performing SDC (Tao et al., 2009; Irfanoglu et al., 2015). These may be incorporated simply by adding similarity terms to the objective function, and may come from different modalities and resolutions. While EPINR's unsupervised nature is appealing, we believe that performance could be greatly improved by incorporating learned priors into the model via methods such as meta-learning (Finn et al., 2017). We would also be interested in expanding our analyses to look at the downstream effects of these SDC methods on microstructure modeling and tractography. Finally, we would also like to expand EPINR to EPI corrections in fMRI, although this task is more challenging as distortion in gradient echo images causes signal dropout instead of "pile-up." If this could be tackled, EPINR would continue to let us to make older data more useful.

## Acknowledgments

We thank Dr. Miaomiao Zhang and the UVA Medical Image Analysis Lab for many helpful discussions. This work was partially supported by NSF Smart and Connected Health grant 2205417.

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

## Appendix A. EPINR Runtime

We note that EPINR does have a high runtime when compared to most other methods. Experiments were run on a workstation with 20 CPU cores and an Nvidia RTX A5000 with 24Gi of GPU memory. ANTs-SyN took approximately 1 minute for one subject, Synb0+topup required around 7 minutes, and QSIPrep took between 60 and 90 minutes to run the pipeline up to the SDC step. EPINR required between 45 and 60 minutes to fit one subject as configured in this experiment. One prominent bottleneck in EPINR's training time is the Hessian calculation for $\mathcal{L}_{\text{bend}}$ which, along with other parameter tweaks, may be disabled for faster training time.

