# OpenReview forum: "EPI Distortion Correction without Opposite Phase Encodings with Unsupervised INR-Based Deformable Registration"
_MIDL.io/2026/Conference — MIDL 2026 Poster_

### Official Review · Reviewer_nDTH · 2026-01-03

**Confidence:** 3
**Preliminary Rating:** 5
**Final Rating:** 5

**Summary:**

The authors tackle the problem of EPI distortion correction in diffusion MRI. Most existing solutions either depend on additional acquisitions or registration on other, unaffected modalities. The authors instead propose to learn the deformation field requires to correction the distortions on a per-image basis using an INR. The authors evaluate their method on multiple datasets, quantifying their results using multiple metrics as well as providing qualitative results.

**Strengths:**

This is a very well written paper with a thorough presentation of the problem at hand, existing solutions and a didactic presentation of their method. The authors evaluate their method against several well established baseline and an ablation version of their method. Multiple datasets are used to assess the performance of their method across several metrics.

**Weaknesses:**

This paper presents very few weaknesses. My main (and minor) criticism would be that the ablation study is somewhat uninteresting. The authors mention that their loss includes a rigid transformation term, but it is omitted halfway into training as it does not do much in the end. A more relevant ablation study would have removed this term from the loss altogether, instead of simply reducing the size of the INR.

As INRs are trained on a per-subject basis, it would have been important to mention the training and inference time, as well as the hardware used to train the network.

Finally, a public implementation would be highly beneficial to the community.

**Detailed Comments:**

An additional criticism: section 4.3 should have come before 4.2, as 4.2 mentions the ablated version of the model before the full model is described.

**Justification Of Final Rating:**

Repeating my comment: the authors have addressed all issues raised during the first round of review. My initial evaluation stands, this is in my opinion a very interesting method and paper. While the training time are certainly concerning and limits real world usage, this is not important enough to warrant a change in score. I have no further comments.

**Justification Of The Preliminary Rating:**

Overall, this is a very strong paper with few errors to address. The proposed method solves a real problem using a clever solution and fewer constraints than existing methods, and performs well. The paper is very well written and guides the reader through all the necessary knowledge to understand the problem and method.

**Questions To Address In The Rebuttal:**

See the weaknesses section and the above comment.

---

> ### Author Response · Authors · 2026-01-25
>
> We thank the reviewer for their generous comments and important critiques. We have taken the provided feedback and improved the manuscript, and we respond to the reviewer's specific comments below.
>
> # Responses to Questions to Address
>
> ## 1. Rigid T1w Ablation
>
> An ablation study was requested by all reviewers, and we agree that this is an important analysis in understanding the EPINR model. We have added an ablation experiment in Section 4.6 that examines the effect of the bending energy regularization, the image Laplacian similarity term, the use of Jacobian modulation during training, the combined regularization terms, and a reduced size network with 128 hidden units and 3 hidden layers.
>
> During this experiment, we found that the learned T1w rigid refinement transformation $A_\rho$ contributed minimally to the final result as our T1w to distorted b0 rigid alignment during preprocessing was highly accurate. So, we have removed this component from EPINR, and all EPINR experiment results have been updated to reflect this.
>
> ## 2. Training Time
>
> We agree, and we have added runtimes and hardware configurations in an appendix. In our experiments, EPINR took between 30 and 45 minutes for one subject, but that can be reduced with different configurations. With 20 CPU cores, Synb0+topup took approximately 7 minutes, and ANTs-SyN took between 30 to 60 seconds, not including rigid pre-registration.
>
> ## 3. Code
>
> We have released a public implementation at <https://github.com/TylerSpears/epinr>, and we have included a link in the manuscript.
>
> ## 4. Section Reordering
>
> We have swapped Sections 4.2 and 4.3 to improve readability.

---

> > ### Comment · Reviewer_nDTH · 2026-01-27
> >
> > The authors have addressed all issues raised during the first round of review.  My initial evaluation stands, this is in my opinion a very interesting method and paper. While the training time are certainly concerning and limits real world usage, this is not important enough to warrant a change in score. I have no further comments.

---

### Official Review · Reviewer_YzoG · 2026-01-07

**Confidence:** 4
**Preliminary Rating:** 3
**Final Rating:** 4

**Summary:**

The authors propose EPINR, an unsupervised Implicit Neural Representation (INR) method for correcting susceptibility distortions (SDC) in diffusion MRI without reverse phase-encoding scans. A SIREN network learns a 1D displacement field constrained to the phase-encoding (PE) direction, mapping a distorted b_0 image to a T1w reference. The continuity and differentiability of SIREN enable analytical calculation of derivatives for regularization and incorporating Jacobian modulation (signal pile-up correction) into the optimization process. While the evaluation on two datasets shows improvements over established tools like ANTs-SyN and Synb0, it lacks in-depth analysis on how and why these improvements are achieved. Furthermore, the novelty seems to be limited to the application of INRs to the problem of SDC, incorporating established constraints and loss functions.

**Strengths:**

* The authors exploit domain specific constraints (1D deformations, Jacobian modulation for signal pile-up) and elegantly integrate these into their framework by exploiting INR specific properties like continuity and differentiability.

* The work targets a widespread issue of correcting legacy dMRI data where reverse-PE scans are missing. Evaluation shows superior performance to established alternative methods.

**Weaknesses:**

* **Limited Novelty:** The paper essentially applies established work on a new problem by performing INR-based multi-modal registration integrating well-known domain-specific constraints like 1D deformation and Jacobian modulation.

* **Lack of Baselines:** The paper lacks comparison to established INR baselines designed for folding-free deformations. For example, recent work by Sideri-Lampretsa et al., “SINR: Spline-enhanced implicit neural representation for multi-modal registration,” proposes a framework more robust to folding. Comparing this work (using the 1D constraint), or methods discussed in the related work, to the proposed framework would help to understand if the analytic calculation of derivatives offers a fundamental advantage over related work.

* **Lack of ablation and critical discussion:**
  * It is unclear how strongly the Jacobian Modulation actually affects the performance in this INR-based setup. For example, the authors claim that it helps to “break through” severe warping; however, there is no specific ablation study to validate the claim.
  * For some cases, the network indicates over-reliance on image similarity (Table 1). EPINR sometimes "outperforms" Ground Truth Topup on similarity metrics, which suggests overfitting to the modality rather than solving the physics problem. Thoroughly analyzing the impact of the proposed similarity terms (MI and NCC) and the balance thereof could provide further insights here.

* **Lack of Runtime Analysis:** Optimization-based INRs can be slow compared to prediction-based CNNs or optimized iterative tools. The paper lacks any runtime comparison.

**Detailed Comments:**

* **Missing INR Baselines:**  It is unclear if EPINR's performance comes from its SDC-specific components (Jacobian modulation, 1D constraint) or simply from using an INR backbone. A comparison to a standard INR registration baseline (like IDIR or SINR constrained to the phase-encoding axis) is missing. This is needed to isolate the specific value of the proposed method.

* **Folding and Regularization:** The authors report 0% negative Jacobian voxels. Recent work (SINR) suggests dense INRs are prone to folding and often need explicit parameterization like B-splines to remain regular. EPINR uses analytic bending energy and smoothness loss. It is not clear if the stability comes from these analytic terms or simply because the 1D constraint restricts the degrees of freedom significantly.

* **Over-reliance on Similarity:** The results show that EPINR sometimes outperforms the Ground Truth Topup on similarity metrics (MI and LNCC). This suggests the network may be overfitting to the T1 reference modality rather than solving the underlying physics problem. A more thorough analysis on whether the model is simply maximizing texture similarity rather than correcting geometric distortion is needed.

**Justification Of Final Rating:**

The authors have thoroughly addressed most of the raised concerns. The revised manuscript, now including an ablation study, provides better insights to the behavior of various proposed components, and more objectively discusses limitations of the work. However, it remains unclear if and when exactly the proposed Jacobian Modulation positively impacts performance. This, together with the long runtimes indicates further potential for future work. Nonetheless, the current work and manuscript can be an interesting  contribution to the conference.

**Justification Of The Preliminary Rating:**

The paper addresses the practical challenge of correcting susceptibility distortions in legacy dMRI datasets without reverse phase-encoding scans. EPINR integrates domain-specific physics (1D constraint, analytic Jacobian modulation) into an INR framework, demonstrating clear performance improvements over established reference-based tools (ANTs-SyN, Synb0).
However, some concerns regarding novelty and validation remain. The method applies existing INR principles without comparing against a standard INR baseline (e.g., IDIR or SINR ), making it difficult to isolate the benefit of the SDC-specific constraints versus the INR backbone. Additionally, the lack of ablation studies for Jacobian modulation and evidence of potential overfitting to the T1 reference (outperforming ground truth on similarity metrics) leave the specific source of improvement unverified.

**Questions To Address In The Rebuttal:**

* **Runtime:** Optimization-based INRs are generally slower than prediction-based CNNs. Please provide the average runtime per subject for EPINR compared to ANTs-SyN and Synb0.
* **Jacobian Modulation Ablation:** The authors claim Jacobian modulation helps "break through" severe warping. Please provide results (quantitative or qualitative) where this term is removed from the loss function to verify if the improvement is really due to physics-informed correction.
* **General INR Comparison:** Why was a general INR registration framework (e.g., IDIR or SINR) not used as a baseline? Comparing against these is necessary to separate the benefits of the SDC constraints from the general benefits of INRs.
* **Physical Plausibility:** In the VCU MS dataset, EPINR has higher Mutual Information than the Ground Truth Topup. This suggests overfitting to the T1 reference. How do you verify the model recovers the true physical geometry rather than just warping the image to maximize texture similarity?
* **Synb0 Comparison:** Synb0 performed poorly, which the authors attribute to resolution mismatches (2.5mm vs native). If Synb0 fails due to simple up-sampling artifacts rather than model capability, the comparison is skewed. Is it possible to retrain Synb0 or to better configure it to the data at hand to ensure a fair comparison?

---

> ### Author Response · Authors · 2026-01-25
>
> We thank the reviewer for their detailed comments and suggestions. We have incorporated their feedback to improve the manuscript, and we respond to their given questions below.
>
> # Responses to Questions to Address
>
> ## 1. Runtime
>
> We have added approximate runtimes in an appendix. As configured in our experiments, EPINR took between 30 and 45 minutes to fit one subject, but that can be reduced with a smaller network, fewer epochs, disabling bending energy regularization, etc. With 20 CPU cores, Synb0 and topup took approximately 7 minutes, and ANTs-SyN took between 30 to 60 seconds, not including rigid pre-registration.
>
> ## 2. Jacobian Modulation Ablation
>
> We have included a general ablation experiment in Section 4.6, and included an ablation of Jacobian modulation. Results are shown in Figure 3 and Table 2. The use of Jacobian modulation during training improved EPINR in the VCU MS dataset, but reduced performance in the MICA-MICS dataset.
>
> We hypothesize that this discrepancy between datasets may be explained by the difference in b0 intensity distributions. As shown in Figure 3 columns 3 and 4, MICA-MICS b0s have distinct boundaries between the gray and white matter, while the VCU MS data is less distinct. This enables some overfitting to the T1w image, which also has distinct gray and white matter intensities. Without Jacobian modulation, the larger distortions in frontal regions are challenging to correct as shown in Figure 3 column 3, and with comparison methods in Figure 2 columns 3 and 4. This prompts some consideration when applying EPINR to different datasets, but we note that EPINR still quantitatively outperforms the comparison methods with or without modulation.
>
> ## 3. General INR Comparison
>
> Our goal with EPINR was to demonstrate how the flexibility and continuous properties of INRs were well-suited to the challenges of SDC in out-of-distribution data, compared to previous SDC methods. This is why we proposed EPINR with a basic SIREN architecture, rather than a more advanced model such as SINR or NePhi. This was essentially the approach with IDIR, and EPINR overlaps heavily with IDIR with SDC-specific constraints. It is very possible that SINR or another architecture would outperform SIREN, but our focus in this work was more on the incorporation of constraints and physical modeling rather than INR architecture. We believe this would be very important to explore in future work.
>
> ## 4. Physical Plausibility
>
> For the VCU MS dataset, we found that both EPINR and Synb0+Topup had higher MI than Topup GT. We also found that EPINR and ANTs-SyN had higher LNCC than topup. This may imply some overfitting to the T1w images, but this would also be implied for Synb0 and ANTs. While the interpretation of this Synb0 result is not obvious, the low parameterization and heavy regularization of ANTs-SyN makes it less likely that T1w texture overfitting is a large factor.
>
> Rather, it seems more likely that the lack of anatomical reference input into topup itself means that topup is just not optimized for matching T1w images. The b0 intensity distribution in MICA-MICS may allow for a high topup MI regardless, but the different intensity distribution of UVA MS means that the baseline similarity will be lower, so there is more opportunity for other algorithms to "outperform" topup. Given this reasoning, we do not believe that a model's anatomical similarity surpassing topup's directly implies overfitting. To address this concern, we have clarified and expanded the discussion on anatomical similarity measures in the manuscript.
>
> We also note that verification of recovering true geometry is indirectly performed by comparing the undistorted b0s and the predicted susceptibility fields to the topup predictions. This is not perfect, of course, but the lack of a real ground truth for real data has been a consistent challenge in the SDC literature.
>
> ## 5. Synb0 Comparison
>
> We believe that there could be several reasons for Synb0's low performance, one of which is the resolution difference from the training data, and another is Synb0's sensitivity to the b0's skull stripping. However, retraining or fine-tuning Synb0 becomes non-trivial for several reasons. First, all methods tested in this work are unsupervised or trained on external datasets (Synb0). Including training data from our datasets would break this unsupervised learning constraint. Second, the native resolutions of MICA-MICS and UVA MS are different (1.6mm and 1.875mm, respectively), and datasets outside of these two will have different resolutions still. So, making Synb0 robust to heterogeneous resolutions would require new multi-resolution training datasets and advanced data augmentation strategies, at the least. At the worst, the Synb0 architecture may need to be updated for generating images at multiple resolutions via upsampling layers or other mechanisms. This is a significant iteration on Synb0, and is out of the scope of this work.

---

### Official Review · Reviewer_LE6j · 2026-01-16

**Confidence:** 3
**Preliminary Rating:** 3
**Final Rating:** 4

**Summary:**

The authors propose to correct the distortion of EPI sequences using only a distorted b0 image and a T1 reference, without relying on opposite phase encoding images, which are not always available in many legacy or clinical diffusion MRI protocols. They build an unsupervised implicit neural representation based registration network that explicitly constrains the deformation to the phase encoding direction and integrates two domain specific priors: Jacobian modulation to model mass preserving intensity changes and analytically computed smoothness and bending energy regularization enabled by the continuous deformation field. Results on two diffusion MRI datasets demonstrate that the proposed method consistently outperforms traditional fieldmap free registration methods and learning based alternatives, and achieves performance close to topup despite not using auxiliary acquisitions.

**Strengths:**

The paper considers a practical and clinically relevant problem of correcting EPI distortion using only b0 images. It further integrates several constraints that are well aligned with the physics of EPI distortion, including restricting deformation to the phase encoding direction, explicitly modeling mass preservation through Jacobian modulation, and enforcing smooth and fold free deformations via higher order regularization.

**Weaknesses:**

1. The paper integrates multiple domain-specific constraints and regularization terms, including Jacobian modulation, Laplacian similarity, smoothness, and bending energy losses. However, there is no ablation study isolating the contribution of each component.
2. On the VCU MS dataset, the topup ground truth produces lower MI and LNCC scores with the T1 reference than the proposed method. Can the authors explain why?
3. The method relies on registration to a T1 weighted anatomical reference, but the paper does not clearly discuss how the T1 is selected or whether multiple T1 acquisitions are possible. It remains unclear how sensitive the results are to the choice of the T1 reference, and whether variations in T1 acquisition could affect the robustness of the proposed approach.

**Detailed Comments:**

N/A

**Justification Of Final Rating:**

Thank the authors for addressing all my concerns and I have raised my score to 4. The additional ablation study clarifies the individual component's contribution clearly. I suggest to explicitly mention that there is a single T1w when describing the dataset to avoid potential confusion.

**Justification Of The Preliminary Rating:**

The paper tackles an interesting and practically important problem, but additional experiments are needed to better validate the proposed approach. In particular, ablation studies isolating the contributions of individual constraints and regularization terms would help clarify which components drive the observed performance gains, and sensitivity analyses examining robustness to the choice of T1 reference would strengthen confidence in the method’s applicability across varied acquisition settings.

**Questions To Address In The Rebuttal:**

1. Include an ablation study that systematically removes or isolates individual loss terms and regularization constraints, such as Jacobian modulation, Laplacian similarity, smoothness, and bending energy.
2. Provide a clearer explanation for why the topup ground truth yields lower intermodal image similarity metrics on the VCU MS dataset, and discuss the limitations of MI and LNCC as evaluation metrics in this context.
3. Clarify how the T1 anatomical reference is selected in practice and evaluate the sensitivity of the proposed method to the choice and quality of the T1 image, for example by testing different T1 acquisitions or perturbations, to demonstrate robustness.

---

> ### Author Response · Authors · 2026-01-25
>
> We thank the reviewer for their time and helpful comments. We have made the necessary additions to the manuscript, and we believe their feedback has improved the submission. We address each of the reviewer's questions below.
>
> # Responses to Questions to Address
>
>
> ## 1. Ablation Study
>
> This was a common request among all reviewers, and we agree that an ablation analysis is important to further validate and clarify EPINR. We have included an ablation experiment in Section 4.6. The quantitative results of this experiment are found in the added Table 2, and example images are shown in the added Figure 3. We performed the ablation experiment on a subset of the full datasets, with 10 randomly selected subjects from each dataset. The results of these experiments indicated that the rigid T1w alignment refinement did not have a strong effect given the highly accurate alignment during preprocessing, and we chose to remove this component from all EPINR experiments.
>
> This new experiment was run on versions of EPINR without: 1) the Laplacian similarity term, 2) the bending energy regularization term, 3) either the bending energy or the Frobenius Jacobian norm regularization terms, and 4) the Jacobian modulation. We also evaluated a smaller variant of EPINR with only 128 hidden units and 3 hidden layers in the SIREN model.
>
> Briefly, this new experiment demonstrated several claims we made in the original manuscript, but with some additional depth. With regards to regularization, we found that the smoothing term was important for reducing overfitting to the T1w and preventing folding in the displacement fields, and the bending energy loss had a small effect in the VCU MS dataset, but a more noticeable effect in MICA-MICS. The Laplacian similarity term also had a small effect in both datasets, but we qualitatively found it helpful for preventing the model from "leaking" b0 warping outside of T1w tissue boundaries. The reduced EPINR 128x3 model had a similar outcome to the previous manuscript's reduced EPINR, where the reduced model outperformed the full in only one metric.
>
> Finally, the usefulness of the Jacobian modulation during training seemed to be dependent on b0 images' intensity distributions, where removing the modulation improved performance in MICA-MICS and reduced performance in VCU MS. We hypothesize that this is dependent upon the level of contrast between white and gray matter in the b0 images, where the higher contrast in MICA-MICS leads to the model "overrusing" the Jacobian modulation to match the contrast levels in the T1w image. This is not as problematic in VCU MS, as the contrast between gray and white matter is very low. As shown in Figure 3 columns 3 and 4, we still see that Jacobian modulation allows EPINR to properly decompress (or re-compress) severe distortions, but at the risk of higher T1w overfitting.
>
> ## 2. Similarity \& Overfitting
>
> We have expanded our explanation of drawbacks in image similarity metrics in Section 4.6. The added ablation experiment demonstrates that these metrics  are susceptible to overfitting. We infer this from the high MI similarity scores for EPINR models without any regularization, which is contrasted by the visible overfitting of the same models in Figure 3 column 2, and the high rate of folding as given by the percentage of negative Jacobian determinants. Overall, we find these similarity metrics to be insufficient when viewed in isolation, with LNCC at least not being directly used during network training. This is not surprising given that these registration models do not rely exclusively on maximizing image similarity. A better comparison could be made between the undistorted b0s and a T2w scan, but this modality was unfortunately not available in all datasets.
>
> ## 3. Anatomical Reference Selection
>
> We have expanded the discussion section to mention the selection of anatomical references for SDC. However, the datasets used in our experiments only provided a single T1w acquisition for each scan, so there was no selection to be made between multiple T1w images. EPINR is flexible in that multiple anatomical acquisitions could be used as part of the objective function, but we considered that as outside the scope of this initial work. As for sensitivity, we note that preprocessing the anatomical references is an important step in using EPINR. We specifically included denoising and debiasing of the T1w and b0 images to simplify the image matching process. These previous algorithms (N4, non-local means filtering, and MP-PCA) have been widely adopted and evaluated, and should help ensure that EPINR performs well in many imaging protocols.

---

### Author Rebuttal · Authors · 2026-01-25

**Rebuttal:**

We thank the reviewers for their time and invaluable feedback. We believe that their comments and questions have improved the manuscript, and we have made the following changes:

1. We have added an ablation experiment in Section 4.6. New quantitative results are shown in Table 2, and qualitative results are shown in Figure 3. We ablated several aspects of EPINR, including the Jacobian modulation, regularization terms, the image Laplacian similarity.

2. We swapped Sections 4.2 and 4.3 for readability.

3. After performing the new ablation analysis, we decided to remove the T1w rigid refinement transformation. This simplifies the model with only a small effect on performance.

4. We have added some discussion regarding multiple anatomical references and objective functions in Section 5.

**Supporting Material:**

/attachment/18a7fa5f9e7fbd582eceaf74747a5e1bac6449a2.pdf

---

> ### Comment · Reviewer_YzoG · 2026-01-27
> **No Highlighted Changes in Revised Manuscript**
>
> Thanks for listing the changes you have conducted in the revised manuscript. Unfortunately, these **changes** do **not** seem to be properly **highlighted** in the provided PDF. If possible, I would ask the authors to fix this for sake of clarity.

---

> > ### Author Response · Authors · 2026-01-27
> >
> > We apologize for the error, we have edited the rebuttal document (with the Program Committee's permission) to now highlights differences in the manuscript. This version is unchanged from the initial rebuttal, beyond the highlighting.

---

### Meta-Review · Area_Chair_XJRN · 2026-02-04

**Recommendation:** Accept (Poster)
**Confidence:** 5

**Metareview:**

All reviewers were satisfied with the rebuttal and reached a consensus on acceptance. EPI distortion correction, in general, is a hard problem to make progress on. The paper tackles a real challenge towards closing the gap to established frameworks such as FSL's TopUp, which requires extra measurements not available for some retrospective analyses.

---

### Decision · Program_Chairs · 2026-02-13

Accept (Poster)